# Structural Relationships between Counselors’ Sense of Calling, Meaning of Work, Living a Calling, and Psychological Burnout

**DOI:** 10.3390/bs14010024

**Published:** 2023-12-28

**Authors:** Shanshu He, Hyesun Kang, Linan Ren, Yiran Li

**Affiliations:** 1College of Administration and Business, Dankook University, 152, Jukjeon-ro, Suji-gu, Yongin-si 16890, Gyeonggi-do, Republic of Korea; heshanshu@dankook.ac.kr (S.H.); renlinan941101@dankook.ac.kr (L.R.); 2School of Management, Kyunghee University, 701, Orbis Hall, 24, Kyungheedae-ro, Dongdaemun-gu, Seoul 02453, Republic of Korea; hsunk@khu.ac.kr; 3Institute for Educational Research, Faculty of Education, Yonsei University, 50, Yonsei-ro, Seodaemun-gu, Seoul 03722, Republic of Korea

**Keywords:** counselor’s calling, sense of calling, meaning of work, living a calling, psychological burnout

## Abstract

This study examined the influence of counselors’ sense of calling on psychological burnout, mediated by meaning of work and living a calling, based on the work as a calling theory (WCT) and preceding studies. Furthermore, the sequential mediating effects of meaning of work and living a calling were investigated. Data were collected from 420 Korean counselors working in counseling centers located nationwide and analyzed using Partial Least Squares Structural Equation Modeling (PLS-SEM). The results revealed that a sense of calling negatively impacted psychological burnout. Second, the sense of calling did not affect psychological burnout through the mediation of meaning of work. Third, the sense of calling negatively impacted psychological burnout through the mediation of living a calling. Fourth, the sense of calling negatively affected psychological burnout through the sequential mediation of meaning of work and living a calling. Based on this study’s findings, implications can be provided to enhance counselors’ professional calling and reduce psychological burnout, thereby aiding them in resolving their psychological issues during counseling practice and providing higher-quality psychological services to clients. Suggestions for improvements and future research are also discussed.

## 1. Introduction

Counselors and mental health professionals play crucial roles in promoting individuals’ mental health and well-being. However, recent preliminary research and reports have indicated that these professionals are increasingly facing psychological burnout [1]. Counselors are at a relatively higher risk in terms of their mental health due to their indirect exposure to traumatic experiences inherent to their job characteristics. Furthermore, when counselors, who play a role in assisting individuals in close proximity, become depleted, it can become challenging to maintain objectivity and demonstrate expertise [2]. Therefore, researching the patterns of burnout among occupational counselors and exploring ways to alleviate it is of utmost importance. This issue manifests itself in different forms and causes across countries worldwide, affecting various aspects of the health of professionals and the quality of psychological services they provide [3,4]. Researchers [1,2] explained that as psychological burnout increases in the workplace, job satisfaction decreases, enthusiasm toward job performance declines, and work effectiveness diminishes. Nam Hyun-Joo and Song Yeon-Joo [5] also reported that when corporate counselors perceive psychological burnout, their psychological health worsens, which negatively affects the quality of psychological services. Similar to this, psychological burnout not only deteriorates the psychological health of counselors but also affects their job efficiency, increases emotional exhaustion, and escalates severe job stress, negatively affecting their developmental competence, which is central to the quality of counseling [6,7]. The objective of this study is to explore conditions that can alleviate psychological burnout among counselors working in South Korea. Specifically, we aim to empirically analyze whether the sense of calling, meaning of work, and living a calling can mitigate psychological burnout among counselors. Particularly, by establishing a sequential mediation model, we intend to reveal the psychological and cognitive processes through which emotional burnout is alleviated.

This approach in our study is grounded in the work as a calling theory (WCT). WCT serves as a useful theory for explaining individuals’ organizational behaviors, positing that a sense of calling can bring about high meaning in work and living a calling, thereby reducing psychological issues [8,9,10]. Duffy et al. [1] proposed the need to scrutinize variables affecting the path from the sense of calling to job satisfaction, considering the developmental nature of a calling and verifying the structural relationship between them. Hence, as part of such validation, this study aimed to explore the structural relationship in which the sense of calling influences the meaning of work, living a calling, and psychological burnout. The perception of a sense of calling has often been suggested as an antecedent that mitigates psychological burnout, with work meaningfulness or the experience of a living calling as mediators. However, previous studies have typically examined the effects of these variables individually. Therefore, in this study, we aim to establish a sequential mediation model, demonstrating the psychological process in which the sense of calling regarding one’s work leads to a greater perception of work meaningfulness and subsequently connects to the perception of living a calling in the job, ultimately alleviating emotional burnout.

### 1.1. Meaning of Work: Antidote to Psychological Burnout

It has been revealed that the meaning of work has a high correlation with people’s prosocial behavior, self-efficacy, and job satisfaction and reduces job stress [11,12]. In particular, studies related to organizational psychology have indicated that the more people understand the meaning and significance of work and tasks in the workplace, the more they can reduce psychological issues or job stress [13]. According to the affective events theory [14], in situations where individuals experience positive emotions, positive attitudes and behaviors are manifested. Accordingly, when individuals experience positive cognitions, such as a sense of calling, they perceive a sense of positive value in the work they are engaged in, leading to a heightened sense of its significance [1,15]. The meaning of work has been identified as a major explanatory variable for job satisfaction, job stress, and psychological issues [16], especially counselors’ psychological burnout [1,17]. Recent studies have shown that meaning of work affects psychological burnout [18].

Hamama-Raz et al. [19] argued that individuals with a high sense of calling experience value and fulfillment in their primary workplace, find meaning in their work, express positive emotions appropriately, and resolve psychological issues by living out their calling as counselors [20]. For example, the higher the sense of calling, the more individuals are engaged in the meaning of calling-related work or their careers [1]; they also focus more on emotions and live out their calling when positive emotions arise in workplace situations and during counseling [21]. While there are limited empirical studies confirming the connection between a sense of calling and the meaning of work, it is reasonable to anticipate that a stronger sense of calling correlates with a greater sense of work’s meaningfulness. That is, the higher the sense of calling, the more likely individuals are to react emotionally to stimuli from calling; even if they experience job stress and psychological distress internally, the possibility of fulfilling a “sense of calling” to oneself or others is high [22]. Therefore, it is likely that individuals with a high sense of calling will appropriately alleviate their negative emotions and express them through the value and fulfillment of their work. In reality, a study showed that among counselors experiencing psychological burnout, those with a high sense of calling increased their occupational self-efficacy and sense of calling and alleviated their psychological issues or severe stress, eventually leading to a decrease in psychological burnout [23]. In this study, based on the arguments and previous research findings of scholars regarding a sense of calling and meaning of work, it was anticipated that the higher the sense of calling, the higher the meaning of work, which will decrease the degree of psychological burnout.

### 1.2. Living a Calling: A Mitigating Factor for Psychological Burnout

The terms “sense of calling” and “living calling” in work are closely related concepts, both centered on the idea of finding a deep sense of purpose or calling in one’s career or work. However, there is a subtle difference between the two. A sense of calling signifies the belief that one’s work or profession plays a central role in the purpose of their life and that their job is being utilized to realize a greater social good [8]. If a sense of calling is defined in this way, then living a calling means believing that performing social virtues through one’s work is being faithfully executed in one’s own life [22]. Existing studies explain that even if one has a sense of calling, its enactment is not automatic, clearly differentiating between the two variables. Specifically, Douglass, Duffy, and Autin [20] emphasized that the benefits of perceiving a calling cannot be fully realized when the degree of living a calling is low. In other words, living a calling should be seen as the key mechanism in the relationship between a sense of calling and positive outcomes such as attitudes and performance.

Recent studies have focused on living a calling as a variable impacting the resolution of psychological issues among counselors [24]. Living a calling is associated with higher levels of job engagement and job satisfaction, lower absenteeism related to job stress, and fewer somatic symptoms [25]. Counselors with a high level of living a calling tend to link their calling with their job performance and counseling tasks to avoid psychological issues or job stress in the workplace [1,11]. Recently, the psychological well-being of counselors has been recognized as vital in delivering effective counseling and psychological services [26]. It is posited that counselors who strongly embody their calling are more inclined to experience a heightened sense of meaning in their work, which could lead to providing higher quality services to clients rather than succumbing to psychological burnout [27].

Living a calling becomes more activated when an individual recognizes the significance and meaning of their work. This activated perception and motivation of living a calling can alleviate emotional burnout through two functions. Firstly, the recognition of life satisfaction through fulfilling a calling helps with focusing less on negative emotions experienced during work [28]. Additionally, actively pursuing a calling inherently enhances job competence, which can reduce job-related stress encountered in the performance of duties. Through these functions, ultimately, the emotional burnout perceived by an individual can be reduced. In fact, counselors who have activated their career development through living a calling tend to perceive lower job stress during the counseling process. Empirical studies exist that suggest this leads to a decrease in psychological burnout [29]. These studies explain that the more counselors engage in living a calling while performing their counseling duties, the more they recognize the value and fulfillment in their work, which ultimately leads to a lower level of psychological burnout.

### 1.3. Research Model and Hypotheses

As explained earlier, the sense of calling refers to a state in which one realizes the social meaning and values associated with their work [1]. It is actively studied as a precursor to psychological burnout [1]. Specifically, when an individual recognizes a sense of calling in their work, they come to realize that they are fulfilling a greater calling in life, which can evoke positive emotions and ultimately lead to a lower level of psychological burnout. Based on these prior studies, we can infer the psychological and cognitive processes that explain why individuals with a high sense of calling experience less burnout in their work. In fact, the work as a calling theory (WCT), which explains the relationship between calling and employee attitudes, posits the meaning of work as a core mechanism [17].

Duffy, Dik, and Douglass [1] presented a WCT model explaining the key mechanism of the positive functions of a sense of calling. Specifically, this model suggests that a sense of calling has positive functions during the process of directly enacting that calling through work. It emphasizes the role of living a calling, activated through a sense of calling rather than the sense of calling itself. Furthermore, the model proposes the meaningfulness of work as a critical variable in the mechanism by which individuals who feel a sense of calling are motivated to actively enact it. It is not just the sense of calling itself but the importance and meaningfulness of the work being carried out due to the sense of calling that encourages active enactment. Specifically, perceiving a calling means that the work identified as a calling takes on greater significance in an individual’s life. Individuals who recognize a higher significance or meaningfulness of their work in their lives are more motivated to live out their calling.

In this way, it explains that people who perceive a calling are more likely to discover a sense of meaning in their work, which, in turn, increases the likelihood of them living out their calling. Existing studies have explored and demonstrated the role of work meaningfulness and a sense of calling as important antecedent variables to reduce emotional burnout. However, research examining the causal relationships between a sense of calling, the meaningfulness of work, and living a calling has been scarce [30]. For instance, Herman et al. [25] pointed out that while a sense of calling, the meaningfulness of work, and living out a calling all show high relevance, the structural relationship between them has not been validated. In response, this study aims to establish a sequential mediation model in which the enhanced sense of work meaningfulness activates living a calling, ultimately leading to decreased emotional burnout.

The research model set in this study is shown in Figure 1. Specifically, to explain counselors’ psychological burnout, sense of calling, meaning of work, and living a calling were set as the main predictive variables. The research hypotheses related to the direct effects are as follows:

**Hypothesis** **1.***The higher the sense of calling, the lower the psychological burnout*.

**Hypothesis** **2.***The negative relationship between a sense of calling and psychological burnout is mediated by the meaningfulness of work*.

**Hypothesis** **3.***The negative relationship between a sense of calling and psychological burnout is mediated by living a calling*.

**Hypothesis** **4.***A sense of calling increases the meaningfulness of work, which in turn enhances living a calling, ultimately reducing psychological burnout*.

## 2. Materials and Methods

### 2.1. Participants and Procedure

This study selected Korean counselors residing in major cities, including Seoul, metropolitan areas, Chungcheong Province, and Jeolla Province, as research participants. Random sampling was also employed through online communities, where most counselors were registered or belonged to online boards of counseling centers. The survey took approximately 15 min to complete, and online gift vouchers were provided as tokens of appreciation to those who completed the responses. Data collection for this research was conducted through online surveys for approximately two months, from 15 February 2023 to 31 March 2023. Although data from 600 counselors were collected, 180 with missing responses for the key variables were excluded, and data from 420 counselors were used in the final analysis. Preliminary statistical power analysis conducted using G*Power3 revealed that for seven measurement variables, a minimum of 153 samples should be collected. This study, which used 420 samples for the final analysis, had a high statistical power [31]. This research obtained approval and authorization from the Institutional Review Board (IRB) of the researchers’ affiliated university, ensuring the scientific validity and reliability of the study. The research was conducted in compliance with research ethics.

The demographic characteristics of the sample are as follows (see Table 1). The significant proportion of non-regular employment in this sample can be attributed to the fact that, in South Korea, individuals working in the counseling profession are more likely to be in non-regular employment compared to regular employment. In South Korea, many counselors are affiliated with private counseling agencies or centers linked to schools and educational authorities, such as Wee Centers. These organizations often tend to employ counselors on a part-time rather than full-time basis. The high proportion of females in the sample can also be attributed to these occupational characteristics.

### 2.2. Measurement

#### 2.2.1. Korean Version of the Working as Meaning Inventory (WAMI)

To measure the meaning of work for counselors, the Korean version of the Working as Meaning Inventory (K-WAMI) was used, which was adapted and validated by Choi and Lee [32] based on Steger et al. [13]. K-WAMI consists of three factors: positive meaning in work, meaning-making through work, and greater good motivation. The scale consists of four items representing positive meaning (e.g., I have found meaningful work (occupation)), three items of meaning-making through work (e.g., I believe the work I do contributes to my personal growth), and three items of greater good motivation (e.g., I know the work I do makes a positive change in the world). The measuring tool was evaluated on a five-point Likert scale (1 = not at all, 5 = very much), and a higher score implied a higher meaning of work. K-WAMI was proven to be valid as the factor loadings for positive meaning, meaning-making through work, and greater good motivation were all above 0.500 in a previous study [32], and the results of the confirmatory factor analysis were also proven to be valid [32]. The meaning of work was significantly correlated with mental well-being (r = 0.510), meaning in life (r = 0.600), and job stress (r = −0.490) [32]. Additionally, the reliability was shown to be 0.730–0.900 in the studies by Kim Soo Jin [33] and Kim Young Ae [34]. In this study, the measurement tool for meaning of work (factor loadings 0.751–0.927) also appeared to adequately measure the latent variables. The overall reliability of meaning of work was 0.830, and the reliabilities of the subscales for positive meaning, meaning-making through work, and greater motivation were 0.843, 0.782, and 0.763, respectively.

#### 2.2.2. Brief Calling Scale (BCS)

In this study, to measure counselors’ sense of calling, we used the Brief Calling Scale (BCS), adapted and validated by Jang and Lee [35] from the original scale by Dik et al. [36]. The sense of calling comprised a single factor with nine items (e.g., I decide on a career path based on the inner needs that guide me. The job I am doing now is my destiny). The measuring tool was evaluated on a six-point Likert scale (1 = not at all, 6 = very much so), with higher scores indicating a higher sense of calling. The validity of the calling scale was supported by previous research, with all observed item factor loadings of 0.500 and above, which were considered valid. Confirmatory factor analysis also showed valid results, with significant correlations confirmed with career commitment (r = 0.680), meaning of work (r = 0.620), and job satisfaction (r = 0.520) [28]. In addition, in a study by Jang and Lee [35], the reliability was 0.970. In this study, BCS (factor loading 0.522–0.790) also showed an acceptable degree of measurement for the latent variable. The overall reliability for the sense of calling was 0.871.

#### 2.2.3. Living Calling Scale (LCS)

To measure counselors’ living calling, this study utilized the Living Calling Scale (LCS) adapted and validated by Jang and Lee [35] from the original scale by Duffy, Allan, and Bott [8] and Dik, Eldridge, Steger, and Duffy [36]. LCS consists of six items (e.g., I often have opportunities to execute my calling; I am realizing my calling in my current job). The measuring tool was evaluated on a seven-point Likert scale (1 = not at all, 7 = very much so), with higher scores indicating more living calling. LCS was measured as a single factor in previous research, with factor loadings between 0.940 and 0.980, which were considered valid. Living calling was significantly correlated with a sense of calling (r = 0.720), work meaning (r = 0.510), life meaning (r = 0.740), job satisfaction (r = 0.570), and life satisfaction (r = 0.720) [35]. In addition, in a study by Jang and Lee [35], the reliability was 0.970. In this study, LCS (factor loading 0.825–0.907) also showed an acceptable degree of measurement for the latent variable with an overall reliability of 0.947.

#### 2.2.4. Maslach Burnout Inventory (MBI)

To measure counselors’ psychological burnout, this study used the Maslach Burnout Inventory (MBI) adapted and validated by Kim Sung Bum [37] from Kang Hak Gu [38], Yu Sung Kyung, and Park Sung Ho’s [39] adaptation and validation of Maslach and Jackson’s [40] psychological burnout scale. MBI consists of nine items on emotional exhaustion (e.g., I feel worn out physically and mentally due to work), five items on depersonalization (e.g., I feel like treating some counselees as if they were inanimate objects), and eight items on reduced personal accomplishment (e.g., I can easily understand what my counselees think and feel), totaling 22 items (with items 4, 7, 9, 12, 17, 18, 19, and 21 being reverse scored). The measuring tool was evaluated on a six-point Likert scale (0 = never, 6 = daily), with higher scores indicating higher psychological burnout. The validity of the psychological burnout scale has been demonstrated in prior research, with factor loadings for emotional exhaustion, depersonalization, and reduced personal accomplishment between 0.830 and 0.870, which are considered atypically valid figures [36,38,41]. Psychological burnout was significantly correlated with surface acting of emotional labor (r = 0.478), deep acting of emotional labor (r = 0.213), emotional dissonance (r = 0.636), and basic psychological need satisfaction (r = −0.459). In addition, in Kang Hak Gu’s [38] study, reliability was shown to include emotional exhaustion (0.857), depersonalization (0.639), and reduced personal accomplishment (0.774). In this study, the psychological burnout scale (factor loading 0.734–0.868) also showed an acceptable degree of measurement for the latent variable. The overall reliability was 0.896, with subscale reliabilities for emotional exhaustion, depersonalization, and reduced personal accomplishment being 0.858, 0.778, and 0.834, respectively.

### 2.3. Data Analysis

In this study, data collected from counselors were analyzed using SPSS 29.0 (demographic characteristics), Amos 29.0 (common method bias post-validation), and SmartPLS 4.0 (structural model, process-adjusted mediation effect). The empirical analysis was divided into measurement and structural models in the following order: First, frequency analysis was performed using SPSS 29.0 to understand the demographic characteristics of the counselors, and reliability and validity analyses were conducted to verify the reliability and validity of the measurement tools used in the measurement model of this study. Second, as the variables of this study were surveyed by targeting a single source, Harman’s one-factor test was performed to verify the error of common method bias. Third, the analysis of the mean, standard deviation, and relationship between variables, as well as discriminant validity analysis, were conducted for the variables utilized in this study. Finally, based on the results of the measurement model, an analysis of the structural equation model (structural model) was conducted to verify the research hypothesis, examining whether counselors’ sense of calling affected psychological burnout through the mediation of work meaning and living a calling. This study used Process Macro based on Partial Least Squares Structural Equation Modeling (PLS-SEM) to verify the sequential mediation model [42]. Unlike the traditional mediation effect analysis method, Process Macro based on PLS-SEM allows the verification of multi-mediation models by verifying one or more mediators at once, reflecting the measurement error of the research model and statistical verification of individual mediation effects [43]. Additionally, it uses Ordinary Least Squares (OLS) regression analysis with the partial least squares method for theoretical exploration [44]. Structural equations using PLS-SEM are characterized by a biased effect estimation compared to structural equations using a correlation matrix (CB-SEM) owing to random measurement errors [45]. However, in the comparative study conducted by Hayes et al. [46], there was no difference in the estimated coefficient values between using OLS regression equations and the Maximum Likelihood (ML) SEM program, even in small samples. There was a difference in standard errors, but since the sample variance estimation of the OLS and ML methods are based on different statistical assumptions, this was expected and, therefore, considered non-problematic.

## 3. Results

### 3.1. Descriptive Statistical Analysis

This part presents the means, standard deviations, and correlations among the latent variables alongside discriminant validity. To ascertain and validate the accuracy of the observational items in depicting the latent variables and secure the discriminative power of the latent variables’ independence, confirmatory factor and discriminant validity analyses were conducted. First, the absolute skewness and kurtosis values of all observational variables did not exceed the respective thresholds of skewness = 2 and kurtosis = 7, which led us to infer that they followed the assumption of a normal distribution [47,48]. Consequently, this study deduced that the assumptions of normality for the structural equation model verification (measurement and structural models) and PLS method were met. The PLS approach considers the collected data as a sample rather than a population, fixes the observations, and draws optimized observations to match the expected values of the estimates with the parameter values, thereby offering a more realistic and unbiased method. As the sample size increases, the estimates approach the actual parameters consistently, and the resulting estimates exhibit similar efficiencies.

Second, in the analysis of convergent validity and cross-loadings, to verify whether the observed items adequately measure and explain the respective latent variables (sense of calling, meaning of work, living a calling, psychological burnout), an analysis of convergent validity and cross-loadings was conducted. The result showed that the measurement model fits the data adequately (R^2^ = 0.444, f^2^ = 0.078 (sense of calling → psychological burnout), Q^2^ = 0.401 (sense of calling, meaning of work, living a calling → psychological burnout)). Unlike the CB-SEM analysis algorithm, PLS-SEM utilizes PLS to analyze structural equations. Using the model fit employed in CB-SEM directly in PLS-SEM may distort the predictive power of the model; hence, R^2^ (explained variance), f^2^ (contribution to explained variance), and Q^2^ (model predictive relevance) were used [44]. Therefore, R^2^ = 0.200 represents weakly explained variance, and R^2^ = 0.500 represents moderate-to-high explained variance; f^2^ = 0.150 represents a weak contribution, and f^2^ = 0.300 represents a strong contribution; while Q^2^ greater than 0 represents ensured predictive relevance, fulfilling the above model fit criteria [44].

Additionally, all observed items were significantly loaded on their respective latent variables (factor loadings β value = 0.522–0.927, *p* < 0.001), indicating that conceptual validity, content validity, and convergent validity were secured [42]. However, this study used the same measurement source to collect data to measure latent variables; therefore, it is not free from common method bias. To address this issue, a post-verification for common method variance, namely Harman’s one-factor test, was performed. By specifying a single factor and conducting exploratory factor analysis, it was found to explain 34.02% of the total variance. In confirmatory factor analysis, the analysis model, which set all observed variables to a single latent factor, showed a significantly lower fit than the measurement model (χ² = 852.264; degrees of freedom: Df = 180; ratio of chi-square minimum and DF: CMIN/DF = 4.735; goodness of fit index: GFI = 0.830; adjusted goodness of fit index: AGFI = 0.782; comparative fit index: CFI = 0.890; normed fit index: NFI = 0.865; incremental fit index: IFI = 0.890; Tucker–Lewis index: TLI = 0.872; root mean square error of approximation: RMSEA = 0.094; root mean residual: RMR = 0.051). Therefore, it was judged that the common method variance problem in the data of the latent variables set in this study was not serious enough to affect the results [49].

Third, correlation and discriminant validity analyses were performed to understand the correlation between the constitutive concepts of the latent variables and the independence of the research variables (sense of calling, meaning of work, living a calling, psychological burnout), indicating the strength of the relationship and ensuring discriminant validity. The analysis of correlation and discriminant validity not only measures the strength of correlation among the main constitutive concepts of latent variables but also observes their relevance before hypothesis testing and ensures discriminative power on the independence of each research variable (sense of calling, meaning of work, living a calling, psychological burnout) used in the study. The relevance and distinction of independent variables, mediators, dependent variables, and moderators were determined and judged by the strength and significance level of the correlation observed among the latent variables, making the analysis of correlation and discriminant validity a prerequisite for hypothesis testing [43]. The correlation analysis results are shown in Table 2.

Sense of calling showed significant correlations with meaning of work (r = 0.64, *p* < 0.000), living a calling (r = 0.78, *p* < 0.000), and psychological burnout (r = −0.54, *p* < 0.000); meaning of work showed significant correlations with living a calling (r = 0.64, *p* < 0.000) and psychological burnout (r = −0.447, *p* < 0.000); and living a calling showed a significant correlation with psychological burnout (r = −0.55, *p* < 0.000).

Discriminant validity was assessed according to the criteria and methods proposed by Fornell and Larcker [42]. Accordingly, the Average Variance Extracted (AVE) of latent variables was compared with the squared value of the correlation coefficients among the concepts of latent variables [42]. If the AVE values of the latent variables were greater than the squared values of the correlation coefficients, there was discriminant validity among the latent variables [42]. As seen in Table 3, the highest correlation coefficient among the variables was 0.78 (between a sense of calling and living a calling), and its squared value, which is the coefficient of determination, was 0.60 (0.78 × 0.78). The smallest AVE value of the latent variables was 0.71, which was higher than 0.60, indicating that the discriminant validity of the latent variables was secured [42]. Composite reliability was determined using AVE and composite reliability (CR), as proposed by Fornell and Larcker [42]. The CR of this study was 0.903 for sense of calling, 0.887 for meaning of work, 0.958 for living a calling, and 0.843 for psychological burnout. This meets the condition of CR with an AVE of 0.500 or higher and a CR of 0.700 or higher [42].

### 3.2. Structural Model and Direct Effects

The structural model in this study appeared to be well-fitted to the data (R^2^ = 0.344, f^2^ = 0.051 (sense of calling → psychological burnout), Q^2^ = 0.401 (sense of calling, meaning of work, living a calling → psychological burnout)). Therefore, the model met the criterion values of model fit, with R^2^ = 0.200 indicating weak explanatory power and 0.500 indicating moderate to high explanatory power; f^2^ = 0.150 indicating weak effect size and 0.300 indicating strong effect size; and Q^2^ being greater than 0, securing predictive relevance [44]. Most of the path coefficients of direct effects were significant; the direct path from a sense of calling to psychological burnout was significant, whereas the direct path from the meaning of work to psychological burnout was not significant. In contrast, the research model explained 34.4% of psychological burnout as a criterion variable. Martens [50] verified the direct effects of the structural model using PLS-SEM to reduce the confirmation bias of respondents. This is because, in situations where theoretical studies are scarce or when the number of research variables increases, researchers use PLS-SEM as an alternative approach to CB-SEM (AMOS) [51]. PLS-SEM not only estimates the relationships of paths with the intention of minimizing the error terms of endogenous constructs in the model but also utilizes available data to estimate coefficients that maximize the explanatory power of endogenous constructs [44,52]. PLS-SEM can select the optimal model through direct trend lines between straightforward and complex models by altering the predictive relationships with the same number of latent variables [44]. Therefore, in this study, the theoretical model was set as the predictive model, and the fit between the models was verified. The research model fitted the data well.

In the final model, the sense of calling showed a significant impact on psychological burnout (β = −0.250, *p* < 0.000). Therefore, Hypothesis 1 has been supported. This finding suggests that counselors with a sense of calling can independently mitigate job stress on their own, resulting in reduced psychological burnout. Sense of calling also demonstrated a significant influence on the meaning of work (β = 0.643, *p* < 0.000), implying that a higher sense of calling leads to a greater sense of value and fulfillment toward one’s work in the workplace. Additionally, sense of calling had a significant effect on living a calling (β = 0.617, *p* < 0.000), indicating that as sense of calling increases, the enactment of living a calling also increases. Living a sense of calling significantly influences psychological burnout (β = −0.301, *p* < 0.000). This implies that engaging in work that one feels called to do can lead to a reduction in perceived job stress, thereby diminishing the likelihood of psychological burnout. However, meaning of work did not show a significant effect on psychological burnout (β = −0.081, *p* > 0.113), indicating that a higher sense of meaning at work does not result in reduced psychological burnout. However, the meaningfulness of work significantly influenced living a calling (β = 0.240, *p* < 0.000), indicating that counselors who perceive their work as meaningful are more inclined to enact their calling.

### 3.3. Mediation Model

To validate the previously established mediation effect hypothesis, PLS-SEM was employed to analyze the structural relationships among the predictor variables, sense of calling, mediating variables, meaning of work and living a calling, and the criterion variable, psychological burnout. A structural model analysis was conducted to examine whether a sense of calling impacts psychological burnout sequentially, mediated by meaning of work and living a calling. Additionally, the Bootstrapping Bias-Corrected Method was used to analyze the mediating effect (sequential mediating effect) of the meaning of work and living a calling. The Bootstrapping Bias-Corrected Method more rigorously reflects the asymmetry of bootstrap estimates compared with the classic statistical estimation method (*p* < 0.05), determining and estimating the validity of the mediation model’s indirect effects through the upper and lower bounds of the confidence interval (modern statistical estimation method) [43]. This method can yield more accurate results when the sample distribution of the estimates is skewed (i.e., not following a normal distribution) [43]. Thus, for indirect effects to be statistically significant in the bootstrapping estimation method, zero should not be included within the 95% confidence interval (between the upper and lower bounds) [43]. This study conducted SEM based on the confidence interval mediation effect estimation method of Hayes [43], and the hypothesis testing results were as follows: the mediation model appeared to be fit for the data (R^2^ = 0.344, f^2^ = 0.051 (sense of calling → psychological burnout), Q^2^ = 0.401 (sense of calling, meaning of work, living a calling → psychological burnout)). As can be seen in Figure 2, sense of calling did not significantly affect psychological burnout mediated by meaning of work (β = −0.052, *p* > 0.118), while it was found to significantly affect psychological burnout mediated by living a calling (β = −0.186, *p* < 0.000). Therefore, the mediating effect of the meaningfulness of work on the relationship between the perception of a calling and emotional burnout was not supported (Hypothesis 2), whereas the mediating effect of living a calling was supported (Hypothesis 3). Second, a sense of calling significantly impacted psychological burnout through a sequential mediation of the meaning of work and living a calling (β = −0.046, *p* < 0.001). As a result, it can be said that Hypothesis 4 has been supported. This indicates that through the meaning of work and living a calling, even when perceiving value and fulfillment in work and experiencing higher job stress, psychological burnout is reduced by enacting a calling (refer to Table 3). Thus, the meaning of work and living a calling mediated the structural relationship between the sense of calling and psychological burnout.

## 4. Discussion

The primary objective of this research is to delve into the correlation between counselors’ sense of calling and psychological burnout. Additionally, we aim to pinpoint the mediating factors involved in these connections, particularly those operating in a sequential manner. Drawing upon pertinent theories such as Duffy et al.’s work as a calling theory [1] and Weiss and Cropanzano’s affective events theory [14], we constructed a sequential mediation model. This model posits that counselors’ sense of calling exerts an influence on psychological burnout by means of mediating variables, specifically work meaningfulness and one’s sense of living a calling. Subsequently, we rigorously tested these hypotheses.

According to the specific results, it was revealed that a sense of calling has a negative impact on psychological burnout through the mediation of living a calling. This implies that counselors who perceive a higher sense of calling tend to engage more in living a calling, thereby acquiring strong vocational values and gratification, which could eventually reduce psychological burnout. These findings partially align with the results of Jang and Lee [35], who indicated that teachers’ sense of calling enhances life satisfaction through the mediation of living a calling.

Second, a sense of calling among counselors did not indirectly influence psychological burnout through the meaning of work. Specifically, the meaning of work did not mediate the relationship between a sense of calling and psychological burnout; however, it was shown to be negatively related to psychological burnout through living a calling. This can be interpreted as follows: even if counselors with a higher sense of calling experience an increased sense of meaning in their work, if they do not carry out this gratification and value by living a calling in their counseling practice, it may not lead to a reduction in psychological burnout. The results of this study, which indicate that the mediating effect of work meaningfulness is not significant when it is introduced alone, suggest that job meaningfulness itself is not a key mechanism in alleviating psychological burnout. The absence of the mediating effect suggests the need to explore other potential mediating variables or pathways.

Furthermore, we analyzed whether a sense of calling can sequentially alleviate psychological burnout through the mediation of work meaning and living a calling, and we confirmed that this relationship is supported. This implies that the positive effects of a sense of calling can lead to desirable outcomes only when they are realized through perceptions of work meaningfulness and actual acts of fulfilling one’s calling. People who have a clear sense of calling are generally more adept at finding meaning in any situation. They tend to interpret their tasks in the context of their calling, making it easier for them to believe that what they are currently doing is meaningful [12]. This heightened perception of job meaningfulness leads to the belief that their work allows them to fulfill their calling, motivating them to more closely align with their sense of calling [1]. For example, a counselor who recognizes the calling to alleviate the suffering of individuals in emotional distress can interpret the simple task of organizing charts in a way that holds greater significance. This interpretation of meaning encourages them to actively seek and execute opportunities to fulfill their calling, even within tasks like answering phone reservations. In this process, individuals can focus less on situations that may lead to burnout while working.

There are numerous studies that focus on discovering a profound meaning in the job related to personal values. However, having a sense of calling goes beyond simply evaluating whether the current tasks have meaning within one’s personal value system; it involves feeling a deeper, almost fateful connection between their job and the individual [17]. As a result, there is a high likelihood of interpreting the meaning inherent in the work more strongly. Moreover, even if individuals recognize a sense of meaning in their work, the positive effects are unlikely to materialize unless they actively engage in behaviors based on such recognition during the task execution process. The results of this study hold practical significance as they provide empirical evidence for reducing psychological burnout among counselors in the counseling process and suggest a more desirable response to job-related stress. According to the results of this study, in order to improve this situation, it is necessary to actively implement opportunities for counselors to practice their calling in the counseling field.

This study’s results have several implications. First, academically, although many studies have been conducted overseas regarding the occupational calling model and emotional event theories, few related studies have been conducted in different cultural contexts. This study, conducted with a sample of Korean counselors, assumes a structural relationship between a sense of calling, work meaning, living a calling, and psychological burnout based on work as a calling model and affective event theory. Its significance lies in elucidating how changes occur in psychological processes (cognition, emotion, attitude, and behavior) within this framework. Furthermore, despite the existence of prior research that has explored the impact of a sense of calling, work meaning, and living a calling on alleviating psychological burnout, there is a scarcity of studies that simultaneously examine the psychological process factors of work meaning and living a calling within the cognitive process. This study was able to advance and differentiate itself from previous research by introducing the psychological processes of work meaning and living a calling into the relationship between a sense of calling and psychological burnout.

The limitations of this study and recommendations for future research are as follows: First, this study included all the data of the counselors in one study, but the aspects of occupational stress experienced by the counselors and the negative outcomes it brings to them might vary due to their age, organizational atmosphere, and so on. If future research is conducted to distinguish age, more implications can be proposed for each age group in the workplace. Second, as this was a cross-sectional study, it was difficult to ascertain the directionality of the variables’ paths. In particular, as the sense of calling may take a long time to show its effect, longitudinal studies with time intervals are needed. Third, it is difficult to conclude that this study is free from common method bias. Future research should explore methods to vary the sources of responses among variables to enhance the study’s objectivity. Finally, it is difficult to definitively say whether the hypotheses were rejected because of the sample size in this study. To enhance the objectivity of the research hypotheses in future research, there is a need to increase the sample size as much as possible and conduct further studies. Nevertheless, this study has contributed to the improvement of the quality of counseling by confirming an overarching model related to alleviating the psychological burnout of counselors who treat numerous psychological distress cases. Future research should continue to develop various models to enhance counselors’ psychological well-being. For instance, counselors who have experienced burnout may not only endure the state of psychological distress but also undergo “post-traumatic growth” as a means to overcome it [53]. In future studies, examining the mechanisms through which counselors overcome burnout resulting from job-related stress can be explored.

## Figures and Tables

**Figure 1 behavsci-14-00024-f001:**
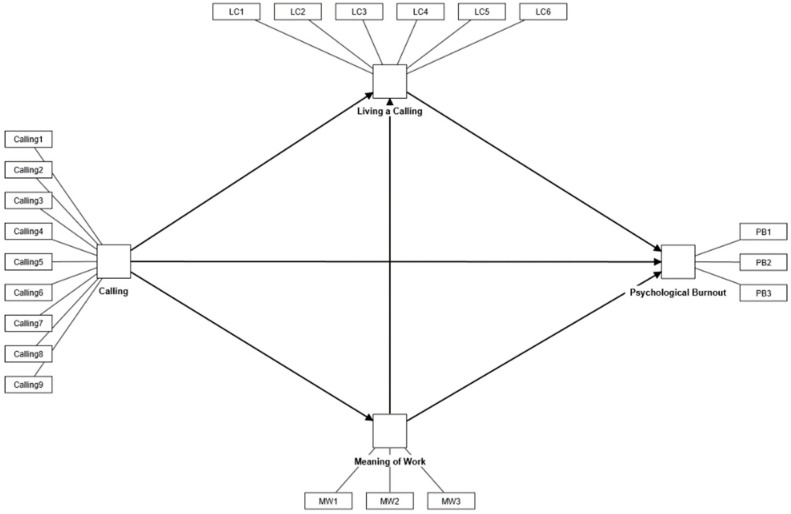
Research model.

**Figure 2 behavsci-14-00024-f002:**
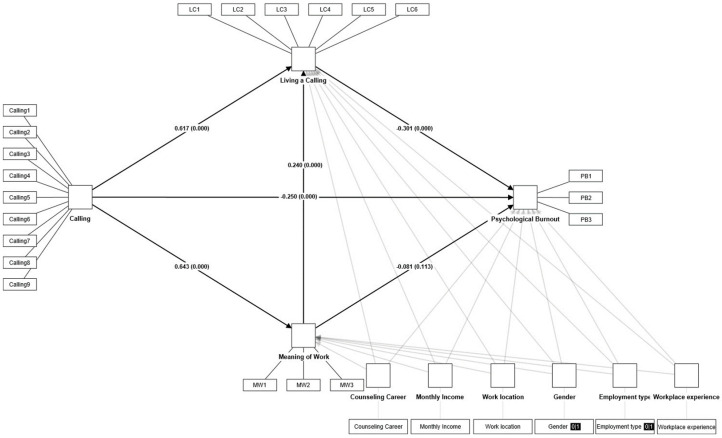
Structural model of sense of calling, meaning of work, living a calling, and psychological burnout. Note. *n* = 420.

**Table 1 behavsci-14-00024-t001:** Demographic characteristics.

Characteristics		Frequency	Ratio (%)
Gender	female	394	93.8
	male	26	6.2
Age	20s	29	6.9
	30s	207	49.3
	40s	138	32.9
	50s	46	11.0
Educational background	bachelor’s degree	2	0.5
	master’s degree	330	78.6
	doctoral coursework	73	17.4
	doctoral degree	15	3.6
Type of institution	youth counseling and welfare centers	68	16.2
	Wee Centers * and classes	41	9.8
	university counseling centers	109	26.0
	private counseling institutions	117	27.9
	corporate counseling institutions	41	9.8
	medical institutions	21	5.0
	research institutions	12	2.9
	religious institutions	11	2.6
Working areas	Seoul City	182	43.3
	small- to medium-sized cities	226	53.8
	rural areas	12	2.9
Employment type	regular employees	126	30.0
	non-regular employees	294	70.0

* The Wee Center is a three-tier integrated student support network involving schools, the South Korean Ministry of Education, and local communities to address difficulties in school life.

**Table 2 behavsci-14-00024-t002:** Mean, standard deviation, correlations, and discriminant validity of measured variables.

Variance	(1)	(2)	(3)	(4)	M	SD	Skewness	Kurtosis
Sense of Calling	**(0.710)**				4.50	0.700	−0.461	0.261
Meaning of Work	0.645 **	**(0.724)**			3.96	0.432	−0.684	1.860
Living a Calling	0.781 **	0.642 **	**(0.793)**		5.09	1.100	−0.424	−0.082
Psychological Burnout	−0.549 **	−0.447 **	−0.559 **	**(0.643)**	2.15	0.730	0.268	0.078

Note: *n* = 420. ** *p* < 0.01. (1) sense of calling; (2) meaning of work; (3) living a calling; (4) psychological burnout. The bold values in parentheses on the diagonal are the values of the Average Variance Extracted (AVE). Below the diagonal are the correlations among the constructs.

**Table 3 behavsci-14-00024-t003:** Verification of indirect effects.

Path	β	SD		T	P	95% CI (LB-UB)
Sense of Calling → Meaning of Work → Psychological Burnout	−0.052	0.033	1.564	0.118	−0.116	0.016
Sense of Calling → Living a Calling → Psychological Burnout	−0.186	0.042	4.458	0.000	−0.270	−0.105
Sense of Calling → Meaning of Work → Living a Calling → Psychological Burnout	−0.046	0.015	3.189	0.001	−0.080	−0.023

Note: *n* = 420. CI, confidence interval; LB, lower bound; UB, upper bound.

## Data Availability

Data collected and analyzed during the study are available upon reasonable request.

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
