# Peer review of "Structural Relationships between Counselors’ Sense of Calling, Meaning of Work, Living a Calling, and Psychological Burnout"

_behavsci, 2023, doi:10.3390/bs14010024_

Round 1
Reviewer 1 Report
Comments and Suggestions for Authors
Dear authors
Thanks for this extensive work
I have some comments about the manuscript that need assessment
1. The introduction is too too lenghty with repition of ideas and it is not organised as you can sense flight of ideas due to this non organization. Please cut it short and explain directly your theory which is actually not well rationalyzed (Sense of calling affects meaning of work which affects living call which affects psychological burnout) as you did not explain the difference between sense of calling and meaning of work they seem very close.
The methodology contain results!?
The results section contains big part of the methodology section
Please rearrange your manuscript and organize it in order to give the reviewer chance to perceive your results and give fair judgment.

Author Response
I would like to express my sincere gratitude for the time and effort you devoted to reviewing this paper and providing invaluable guidance. Your guidance has played a pivotal role in enhancing the quality of this paper.
To better convey the research's purpose, content, and significance, I have made substantial revisions to both the introduction and the entire text.
Specifically, I have presented demographic characteristics in a table within the methodology section to reduce confusion. Furthermore, I have improved Table 2 (Table 1 in the previous document) to enhance the understanding of the analysis results. These changes have resulted in a clearer and more precise description of the research methodology.
Throughout the document, I have also taken measures to improve readability by ensuring consistency in terminology and eliminating repetitive or redundant elements.
Once again, I deeply appreciate your invaluable insights and your careful evaluation of this paper.

Reviewer 2 Report
Comments and Suggestions for Authors
This study looked at several self-reported variables capturing meaning at work and sense of calling with regards to psychological burnout among counselors in Korea. The authors used structural equation modeling to explore indirect pathways. The data was based on cross-sectional surveys. This manuscript lacked clarity in many regards and would need to be improved in order to be considered for publication. Below is a summary of some of the areas where clarity could be improved.
A lot of different terms are used but not explained. What is professional burnout compared to psychological burnout? What are corporate counselors? Why is “sense of mission” sometimes used as well as “sense of calling”? Is there a difference between these concepts? The term psychological burnout is used throughout the manuscript, but then suddenly the term “psychological exhaustion” is used in the research model and hypothesis section.
The introduction is very long, and the hypotheses are numerous, but the authors do not clearly outline what the main concepts are and how they are expected to relate to each other. Perhaps it would be easier to follow the introduction first introduced and defined the main concepts, introduced the theoretical model, described the aim and hypotheses, then described the previous literature.
An aim is introduced on page 2, but I think it is rather misplaced and doesn’t really describe the study that is actually done. “In this study, the psychological health issues faced by counselors were conceptualized as psychological burnout to not only examine the developmental competence of counselors, but also to examine the relationship of sense of calling, meaning of work, and living a calling with psychological issues such as burnout. This study aimed to identify the variables that intensify this relationship.” First of all, only one outcome is analyzed so I think the use of the term “such as” is misleading here. Second of all, I think looking at variables that “intensify” the relationship would be more of an analysis of effect modification or interaction which this study does not do.
The introduction does not adequately describe the motivation for doing the study and the knowledge gap that the study fills. An example is the sentence on page 2 “In particular, the meaning of work and living a calling are known to have a close relationship with negative variables of psychological burnout through sense of calling.” If this is known, then what is the point of the study? The discussion mentions that this has not been investigated in a “domestic” context, but none of this is justified in the introduction.
There are some hypotheses mentioned throughout the introduction when summarizing the literature before all hypotheses are introduced in section 1.3. The authors could either gradually introduce the hypotheses in each section or introduce them all in one section, but it is unnecessary to do both.
The long list of hypotheses does not really reflect what is reported in the results section of the hypothesis. I think all of this could be simplified to a few key hypotheses which are tested and reported.
The figures are messy and not possible to understand on their own.
The methods say that individuals were targeted based on having a sense of calling. Wouldn’t this be a major source of selection bias which makes it impossible to make comparisons based on sense of calling?
What do the authors mean when they say that they removed “insincere” responses?
What are Wee centers? What is meant by “regular employment”? (page 7)
Why not have a table summarizing the demographic information rather than describing it all in text?
It is really difficult to distinguish the difference between sense of calling and living a calling. It seems like these concepts overlap quite a bit. Could this be made clearer?
There are words missing in the first sentence of the results.
Furthermore, the results section contains a combination of methods, results, and interpretations and should only present the results.
Table 1 is extremely difficult to read. All the constructs on the top row blend together and are not separated. The average variance extracted values presented alongside the correlation coefficient is confusing. Couldn’t the authors remove the psychological burnout column and put a separate column for the AVE values?
If this article is targeted to an international readership, the term “domestic” should not be used, but rather the country where the research was done. Furthermore, why is the context important here? What is different in this context which would require research in this context?
Finally, not being able to confirm a hypothesis is not a “limitation” in itself. Why would we do research if we already knew the outcome?
Author Response
I sincerely appreciate the time and effort you dedicated to reviewing this paper
and providing your invaluable guidance.
Your guidance has truly been instrumental in improving this paper.
Below, I have outlined the areas where I have incorporated the revisions you suggested.
Once again, I extend my gratitude for your advice for this paper.
- What is the difference between professional burnout and psychological burnout?"
: I have conscientiously reviewed the entire document and made the adjustment of replacing the term 'professional burnout' with the phrase 'psychological exhaustion' throughout.
- The description of corporate counselors is insufficient.
: I have thoughtfully included an explanation of the counselor within the document.
- What is the difference between a 'sense of mission' and a 'sense of calling'?
: To ensure clarity of meaning, I have consistently used the term "sense of calling" throughout the entire document.
- What is the difference between psychological burnout and psychological exhaustion?
: In order to reduce confusion, I have standardized all terminology within the document to "psychological burnout."
- The key concepts and hypothesis relationships, introduction of the theoretical model, research objectives, and the introduction need to be revised overall.
: I am grateful for your guidance, and I have made revisions to the introduction, key concepts and relationships introduction, as well as the model introduction, taking your advice into consideration.
- The description of the research content needs to be revised.
: I have reviewed and revised the entire text to ensure that the content is conveyed more clearly.
- The introduction should include the academic significance of how this study complements existing research.
: Incorporating your valuable advice, I have introduced the sequential mediation model in the introduction, highlighting it as the distinguishing feature of this study. This model serves to elucidate the mechanisms explaining the relationship between independent and dependent variables and has been empirically validated using data.
- The hypotheses need to be revised.
: I have meticulously described our hypotheses, clearly outlining them as four distinct ones. Furthermore, I have removed any previously confusing descriptions to enhance clarity.
- An explanation is needed for the deletion of insincere responses.
: I have removed the data that lacked information for the key variables and included this explanation for clarity.
- An explanation is needed for Wi-Center, and the distinction between regular and non-regular employment.
: I have added an explanation regarding the term "Wee Center," which refers to student support facilities in South Korea. Additionally, I have included an explanation for the high proportion of non-regular employees in the demographic characteristics of the sample.
- Present the demographic information in a table.
: I have removed the textual description of demographic information and presented it in the form of a table for clarity.
- Distinguish between a sense of calling and living a calling.
: I have added a detailed explanation to clarify the differences between a sense of calling and living calling.
- Distinguish between methods, outcomes, and result.
: I have made revisions to the description of the research methodology section, including the presentation of results, to enhance clarity. Additionally, I have improved Table 2 (formerly Table 1 in the previous document) for better readability.
- Exclude the term "domestic."
: Within the document, I have removed the term "domestic" and instead provided explanations using "Korea."
-The end-

Round 2
Reviewer 1 Report
Comments and Suggestions for Authors
Still I insist that the results do not reflect the conclusion and the mediated process is not well explained and presented to be accepted and even the results do not support the conclusion.
there are mainly paragraphs that contradict each other in the paper for example from line 70 to line 85
The introduction still is too too lengthy unnecessarily .
Comments on the Quality of English Languageneeds minor revision
Author Response
I am deeply grateful for the time and effort you invested in critiquing this paper and offering your invaluable insights. Your advice has been fundamentally important in enhancing the quality of the paper. In the following,
I have detailed the sections where I have implemented the changes you recommended.
- Still I insist that the results do not reflect the conclusion and the mediated process is not well explained and presented to be accepted and even the results do not support the conclusion.
: We appreciate your feedback on the conclusion and presentation of the mediated process in the study. In response, We have provided a more detailed step-by-step explanation of each stage of the mediation process, including specific aspects that were not clearly explained before. In addition, We have further strengthened the practical implications throughout the conclusion.
- Regarding the conciseness of the introduction and revision of some paragraphs...
: We have revised the sections in the introduction (from line 70 to line 85) where the message was described ambiguously, ensuring they are now clearer and more understandable. Additionally, We have rewritten parts that were unnecessarily detailed to present the introduction more succinctly.
- Comments on the Quality of English Language needs minor revision.
: We have carefully reviewed the entire document once more to ensure that there are no issues with the English language usage throughout.
I would like to reiterate my thanks for your valuable input on this paper.
-The end-
Reviewer 2 Report
Comments and Suggestions for Authors
The authors have done a good job editing the manuscript based on my suggestions.
Author Response
Thank you very much for taking the time to review our paper. We are grateful for your insights which have helped to improve its quality. Please take care of your health. Have a great day.
Round 3
Reviewer 1 Report
Comments and Suggestions for Authors
Dear authors
My final advice to you is to read your paper thoroughly yourself by the the most senior of you and analyze each sentence.
In this study, we aim to explore conditions that can alleviate 49
emotional burnout among counselors working in South Korea. Spe-cifically, we seek to 50
empirically analyze whether the perceiving a calling, the mean-ingfulness of work, and 51
the living a calling can mitigate emotional burnout in counse-lors, and if so, through what 52
mechanisms
Do you think this objective is coherent?
You are looking for factors that alleviate burnout and in the next sentence you mention that you want to study if living call mitigate burnout!!!
Comments on the Quality of English Language
Secondly, I do not understand why did you break most of the words in your paper , for example (counse-lors, mean-ingfulness, Spe-cifically)
Do you think this is an appropriate way of writing English language? your paper is full of this
Author Response
I am deeply grateful for the time and effort you invested in critiquing this paper and offering your invaluable insights. Your advice has been fundamentally important in enhancing the quality of the paper. In the following,
I have detailed the sections where I have implemented the changes you recommended.
- My final advice to you is to read your paper thoroughly yourself by the the most senior of you and analyze each sentence.
In this study, we aim to explore conditions that can alleviate emotional burnout among counselors working in South Korea. Spe-cifically, we seek to empirically analyze whether the perceiving a calling, the mean-ingfulness of work, and the living a calling can mitigate emotional burnout in counse-lors, and if so, through what mechanisms
- Do you think this objective is coherent? You are looking for factors that alleviate burnout and in the next sentence you mention that you want to study if living call mitigate burnout!!! Secondly, I do not understand why did you break most of the words in your paper , for example (counse-lors, mean-ingfulness, Spe-cifically) Do you think this is an appropriate way of writing English language? your paper is full of this
Based on the above feedback, I have revised the entire paper. The revised sections have been marked in red. Thank you.
I would like to reiterate my thanks for your valuable input on this paper.
-The end-